# Does quality of life return to pre-treatment levels five years after curative intent surgery for colorectal cancer? Evidence from the ColoREctal Wellbeing (CREW) study

**Sally Wheelwright[1], Natalia V. Permyakova[1], Lynn Calman[1], Amy Din[1], Deborah Fenlon[2], Alison Richardson[3,4], Samantha Sodergren[1], Peter W. F. Smith[5], Jane Winter[1,4], Claire Foster[1]\*, Members of the Study Advisory Committee[¶]**

1 Macmillan Survivorship Research Group, School of Health Sciences, University of Southampton, Southampton, United Kingdom, 2 College of Health and Human Sciences, Swansea University, Swansea, United Kingdom, 3 University Hospital Southampton NHS Foundation Trust, Southampton, United Kingdom, 4 School of Health Sciences, University of Southampton, Southampton, United Kingdom, 5 Social Statistics and Demography, Social Sciences, University of Southampton, Southampton, United Kingdom

¶ Members of the Study Advisory Committee is listed in the Acknowledgments.
* C.L.Foster@soton.ac.uk

**Data Availability Statement:** The data underlying the results presented in the study are available

## Abstract

### Background

The ColoREctal Wellbeing (CREW) study is the first study to prospectively recruit colorectal cancer (CRC) patients, carry out the baseline assessment pre-treatment and follow patients up over five years to delineate the impact of treatment on health and wellbeing.

### Methods

CRC patients received questionnaires at baseline (pre-surgery), 3, 9, 15, 24, 36, 48 and 60 months. The primary outcome was Quality of Life in Adult Cancer Survivors (QLACS); self-efficacy, mental health, social support, affect, socio-demographics, clinical and treatment characteristics were also assessed. Representativeness was evaluated. Predictors at baseline and at 24 months of subsequent worsened quality of life (QOL) were identified using multivariable regression models.

### Results

A representative cohort of 1017 non-metastatic CRC patients were recruited from 29 UK cancer centres. Around one third did not return to pre-surgery levels of QOL five years after treatment. Baseline factors associated with worsened QOL included >2 comorbidities, neoadjuvant treatment, high negative affect and low levels of self-efficacy, social support and positive affect. Predictors at 24 months included older age, low positive affect, high negative affect, fatigue and poor cognitive functioning.

from http://www.horizons-hub.org.uk/access_data.html.

**Funding:** The study was funded by Macmillan Cancer Support as part of the Macmillan Survivorship Research Group programme (Chief Investigator: CF). SW, NP, LC, AD, SS, CF were funded by the Macmillan Survivorship Research Group. DF was funded by the Macmillan Survivorship Research Group during recruitment for CREW and subsequently by University of Southampton and then Swansea University. AR and PWS are funded by the University of Southampton. JW is funded by University Hospital Southampton NHS Foundation Trust. Macmillan Cancer Support had no role in study design, data collection and analysis, decision to publish, or preparation of the manuscript.

**Competing interests:** I have read the journal's policy and one author of this manuscript has the following competing interest: Professor Deborah Fenlon has received an honorarium for teaching from Roche. This does not alter our adherence to PLOS ONE policies and sharing data and materials.

## Conclusions

Some risk factors for poor outcome up to five years following CRC surgery, such as self-efficacy, social support and comorbidity management, are amenable to change. Assessment of these factors from diagnosis to identify those most likely to need support in their recovery is warranted. Early intervention has the potential to improve outcomes.

## Background

With an estimated five-year prevalence of 3.5 million worldwide, survivors of colorectal cancer (CRC) form the largest group of cancer survivors affecting both men and women [1]. Improvements in early diagnosis and treatment mean that in England, for example, five year survival rates are approaching 60% [2]. Although survival rates continue to improve, knowledge about recovery from CRC in terms of health and wellbeing is limited because most previous studies have either been cross-sectional or have not assessed health and wellbeing before treatment [3, 4]. This means that individual recovery over time cannot be explored. In order to identify factors that might predict who will do poorly after treatment, or benefit from additional support, it is important to understand this population and their health at cancer diagnosis.

Another key point in the cancer journey is when regular oncology/surgical appointments come to an end. Although some people will move to self-managed follow-up a few weeks after surgery, they will be invited to regular surveillance appointments over the following months, before the interval between these appointments gradually increases to years. This period of time, as contact with specialists reduces and when people start to adjust to life after treatment, can be a source of distress for some people who have reported no or minimal distress previously [5]. In the UK for example, typically, after the two year surveillance appointment, the next appointment will be the five year surveillance appointment [6]. The two year surveillance appointment provides an opportunity to identify individuals at risk of poor QOL outcome and establish care plans for people who may not be seen again in secondary or tertiary care for three years.

Previous large-scale studies have suggested that some CRC survivors fare reasonably well with respect to long term health and wellbeing outcomes. For example, a population-level study in England of individuals diagnosed with CRC 12–36 months previously found, using the EQ-5D [7], more than one third reported perfect health [8]. In Australia, there was a statistically significant improvement in mean QOL, as assessed by the Satisfaction with Life Scale [9], between five months and five years after diagnosis [3, 10]. Whilst in Germany, health-related QOL, as assessed by the European Organisation for Research and Treatment of Cancer Quality of Life Core Questionnaire (EORTC QLQ-C30) [11] 1, 3, 5, and 10 years after diagnosis in a population-based CRC cohort, was comparable or better in older survivors than controls up to five years after diagnosis (although the same was not true of younger survivors and the results were less favourable for older survivors 5–10 years after diagnosis) [12]. Similarly, in the Netherlands, there were no differences on the EORTC QLQ-C30 between people aged ≥70 years diagnosed with CRC up to ten years earlier and an age and sex-matched normative sample. Although people aged <70 years in the CRC group had more symptoms and functioning problems than the age and sex-matched normative sample, there was no difference on global QOL [13].

Despite some people recovering well after treatment for CRC, we have previously demonstrated that around one third of individuals will experience significant recovery problems or may fare consistently poorly during recovery, with pre-treatment psychosocial factors, independent of treatment or disease characteristics, predicting wellbeing outcomes up to two years following surgery [14]. The problems associated with CRC and its treatment, such as fatigue, psychological distress, sexual dysfunction and altered bowel habits, may be long-lasting and can have an impact on physical functioning, making it difficult to return to work or resume the activities enjoyed prior to diagnosis [15]. There may be additional problems for those patients living with a stoma. These persistent difficulties alongside continued feelings of uncertainty and concerns for the future not only have implications for an individual's health but also their sense of subjective wellbeing.

The ColoREctal Wellbeing (CREW) study [16] was established to systematically follow a representative cohort of patients from diagnosis using a framework of recovery of health and wellbeing [17] to explore the multiple factors that might affect recovery over time. Unlike other longitudinal cohort studies of CRC, the first assessment was carried out pre-treatment. The primary outcome measure was the Quality of Life in Adult Cancer Survivors Scale (QLACS) [18]. This measure was selected over other cancer-specific quality of life instruments because it includes longer term issues, rather than focussing on the acute effects of diagnosis and treatment.

The aim of this paper is to describe the CREW cohort in terms of socio-demographic, clinical and treatment characteristics and to report the results of the primary outcome (QLACS), up to five years post-surgery. The objectives are: i) to evaluate the representativeness of the CREW sample, both in terms of how the cohort compares with UK national data and also by comparing the participants who remained in the study throughout the five years with those who did not; ii) to identify the proportion of CREW participants whose QOL improves, declines or stays the same compared with baseline at each time point; and iii) to determine which characteristics at diagnosis and at two years are associated with a subsequent decline in QOL.

## Methods

### Procedure

The study was approved by the UK National Health Service National Research Ethics Service (REC reference number: 10/H0605/31) and all participants provided written informed consent. Some patients gave consent to be followed up but not to complete the self-report questionnaires–reduced consenters (Fig 1).

All eligible patients attending 29 UK cancer centres during the recruitment period (November 2010 –March 2012) were invited to participate. Not all centres recruited throughout the recruitment phase and recruitment closed when the target sample of 1000 patients was reached.

Baseline questionnaires were completed prior to primary surgery wherever possible, then follow-up questionnaires mailed to participants at 3, 9, 15, 24, 36, 48 and 60 months. Before each mailing, general practitioners (GPs) were contacted to check that participants were alive and had mental capacity. Reminders by phone or post were issued two and four weeks later where necessary. From 15 months onwards, a short version of the questionnaire was included in the reminder pack as an alternative to filling out the full version (see Supporting information for differences between the two). Participants who were not sent a questionnaire at one time point remained eligible for the next questionnaire.

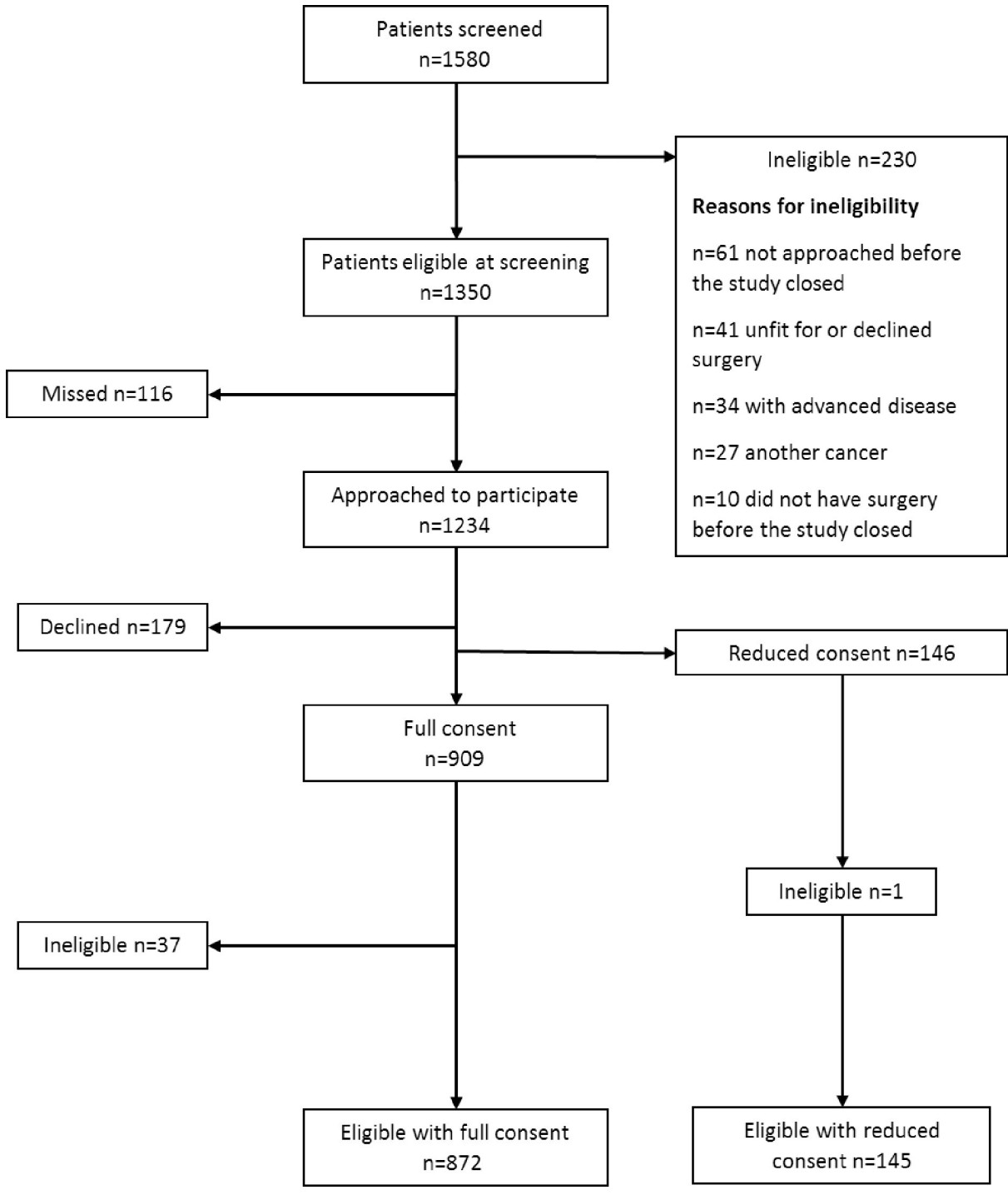

**Fig 1. CREW recruitment flowchart.**

In order to assess potential selection bias, non-identifiable socio-demographic data (with consent) were collected (age, gender, ethnicity, domestic status, employment status) on those who chose not to participate in the study. Where possible, reasons for non-participation were gathered.

Ethnicity was self-reported by full consent participants or by the research nurse for those who gave reduced consent. Postcode was used to derive a measure of neighbourhood deprivation (Index of Multiple Deprivation) [19]. Clinical and treatment details were extracted from medical notes at 6 and 24 months after surgery.

## Participants

Patients were eligible if they: a) had a diagnosis of colorectal cancer (Dukes' stage A-C), b) were awaiting primary surgery with curative intent, c) $\geq$ 18 years old, d) had the ability to complete questionnaires (language line translation facilities were available for those that did not speak English). Distant metastatic disease at diagnosis or a prior diagnosis of cancer (other than non-melanomatous skin cancer or in situ carcinoma cervix) were exclusion criteria.

## Measures

Selection of measures was based on our recovery framework [17]. Table 1 lists the measures used, the area of assessment and the relevant recovery framework domain [17]. Supporting information details which measures were included at each time point. The primary outcome, QLACS, comprises 28 generic items, with seven domains (negative feelings, positive feelings, cognitive problems, sexual problems, physical pain, fatigue and social avoidance) and 19 cancer-specific items, with five domains (appearance concerns, financial problems, distress over recurrence, family-related distress and benefits of cancer). A generic summary score (GSS) is created by summing the generic domains (after reverse scoring the positive feelings domain). A cancer specific summary score (CSS) is created by summing the cancer-specific domains, except benefits of cancer. Higher scores indicate worse QOL. The 28 generic items were included at each time point and the 19 cancer-specific items were included from 15 months onwards, as these are more relevant to longer term survivors. This paper reports on the GSS.

Further details about the other measures have been reported elsewhere [14, 16, 20].

## Statistical methods

Statistical significance was set at 5%. Due to skewed distributions, some covariates were categorised using published clinically significant cut-offs: Personal Wellbeing Index (PWI) (cut-

**Table 1. Validated measures used in CREW.**

| Framework domain | Measure |
|---|---|
| Problems experienced | EORTC QLQ-C30 [11] and QLQ-CR29 [21] |
| | Supportive care needs survey (SCNS-SF34) [22] |
| Health and wellbeing | Quality of life in adult cancer survivors (QLACS) [18] |
| | Personal wellbeing index-adult (PWI-A) [23] |
| | EQ-5D [7] |
| Pre-existing factors | List of threatening experiences [24] |
| Personal factors | Illness perception questionnaire (revised), IPQ-R [25] |
| | Self-efficacy for managing chronic disease, SEMCD [26] |
| | Cancer survivor self-efficacy scale, CS-SES [27] |
| | PANAS scale, PANAS [28] |
| | State trait anxiety index, STAI [29] |
| | Centre for epidemiological studies depression, CES-D [30] |
| Environmental factors | Medical outcomes study social support survey, MOS-SSS [31] |
| Coping and self-management | Brief cope inventory, COPE [32] |

off = 70) [23], STAI (cut-off = 40) [33] and Centre for epidemiological studies depression (CES-D) (cut-off = 20) [34, 35]. Scores on the Cancer Survivors' Self-Efficacy Scale (SEMCD/ CS-SES) were categorised as very confident, confident, moderately confident and low confidence [20], and scores on the Modified Social Support Survey (MOS-SSS) were divided into ceiling (feeling fully supported) and below.

Descriptive analyses compared baseline characteristics by consent status (full versus reduced) and, full consenters' 5-year follow-up status (participated at 5 years versus did not, excluding deceased) using chi-squared or chi-squared test for trend, as appropriate. Additionally, a multivariable logistic regression model was constructed to compare the baseline measures of participants who did/did not return questionnaires at 5 years.

Mean values of the QLACS-GSS were compared between the time points using a t-test. Changes in QOL for each individual were defined using the QLACS-GSS based on the rough rule of thumb for QOL instruments that a clinically significant difference is indicated by a 10% change in score [36]. As a higher score represents worse QOL on the QLACS, participants were classified as "improved" if their QLACS-GSS was <90% of their baseline score; "worse" if their QLACS-GSS was >110% of their baseline score; otherwise, "same". The proportion of participants in each category over follow up was calculated.

Two multivariable regression models were produced with worse QOL as a time-varying outcome: model one examined the associations between participant characteristics at baseline and worsened QOL up to 2 years after surgery; model two examined associations between participant characteristics at 2 years post-surgery and worsened QOL over the next 3 years. The variables included in each model are listed in supporting information. For both models, a binary outcome indicating same/better (= 0, reference) or worse (= 1) QOL was calculated, by comparing the QLACS-GSS for subsequent time points with the QLACS-GSS at the time point of that model (model one at baseline, model two at 2 years). The population-average approach was used to combine data for each individual across the required time points for each regression model. Standard errors were adjusted to account for repeated observations of the same individuals. Elimination of insignificant covariates was performed with the backwards selection approach using Stata 14 software.

## Results

### Participants

Figs 1 and 2 show the flow of patients through the study. All patients presenting with colorectal cancer in the timeframe of the study were screened. Reasons for ineligibility at screening are shown in Fig 1. Some patients were found to be ineligible post-surgery due to advanced disease or not having cancer. Reasons for declining to take part included anxiety about treatment, not wanting to be followed-up for a long period, too ill to complete questionnaires or already in a study. The final total was 1017 (872 participants full consent; 145 reduced consent).

Demographic profiles of decliners and consenters were broadly similar. Of the 179 eligible individuals who declined, the mean age was 72.1 years (SD = 9.8), 43.8% were male and 36.5% female (19.7% not recorded), compared with an average age of 68.7 years (SD = 10.7) and 58.7% males for the 1017 consenters. There were similar distributions of ethnicity and domestic status in the decliners and consenters.

As 10 participants withdrew consent and one person died at baseline, socio-demographic data are reported on 1006 participants. Clinical and treatment information is presented for 997 participants (4 participants did not consent to the collection of medical details, 1 changed hospitals before details could be collected and a further 4 were missed). Response rates remained high throughout the study, ranging from 71 to 88%.

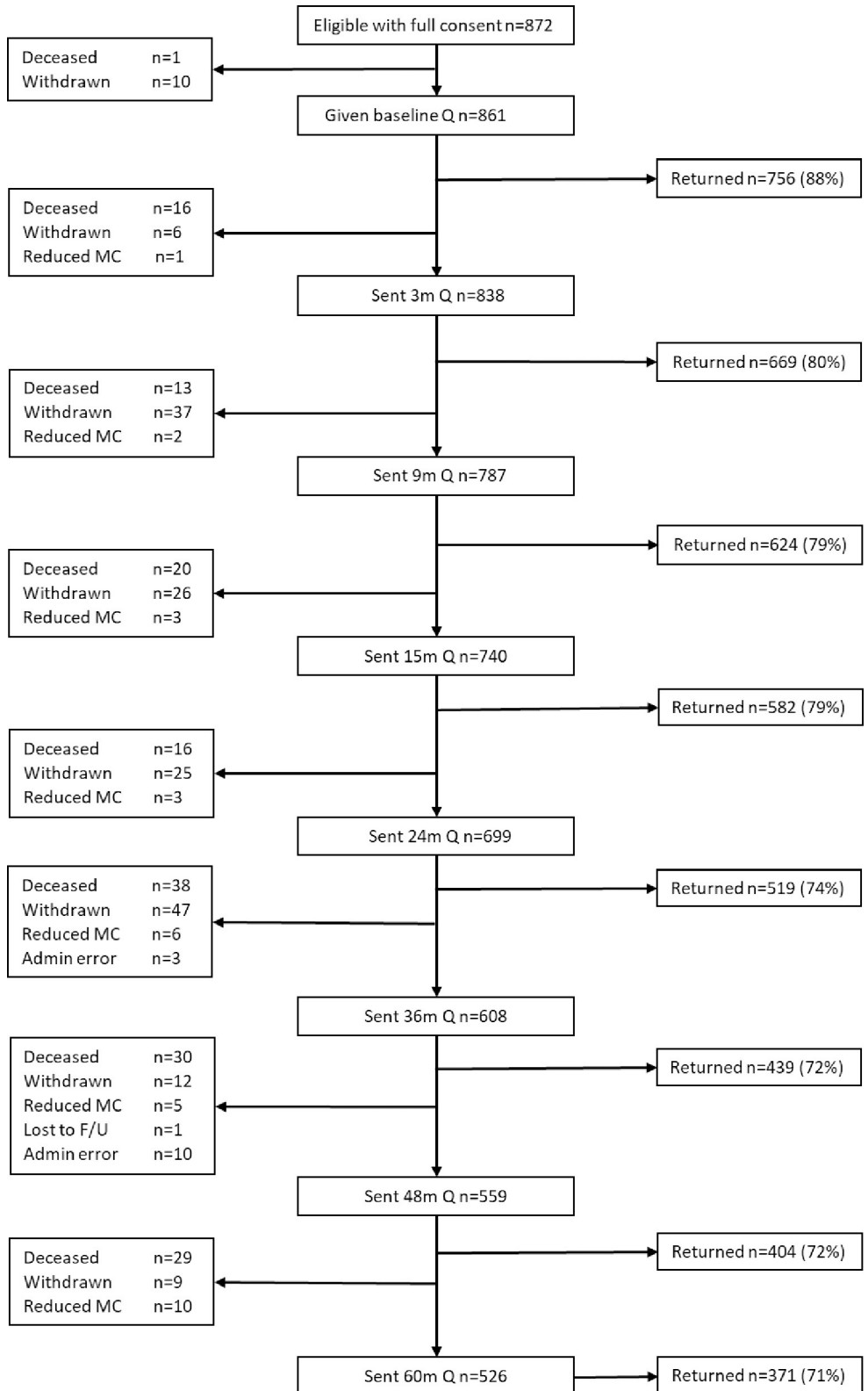

**Fig 2. CREW participation flowchart.** Abbreviations: Follow up (F/U), Mental capacity (MC), Questionnaire (Q). Participants who were not sent a questionnaire because of mental capacity issues or through administrative error remained eligible for the questionnaire at the next time point.

Most participants were from England (85.6%), with 10.9% from Wales and 3.5% from Scotland. Socio-demographic characteristics are presented in Table 2 and clinical and treatment characteristics in Table 3. These characteristics were broadly comparable to the contemporary National Bowel Cancer Audit (NBOCA) dataset [37]; although direct comparison was not possible as Dukes' stage D patients were excluded from our cohort whilst the NBOCA reports aggregate results for patients with Dukes' stage A-D. Hence in CREW, there were more screen-detected and fewer emergency surgery presentations compared with national statistics. The other main difference between CREW and national datasets was that many more CREW participants received neo-adjuvant radiotherapy for rectal cancer (48% in CREW vs 11% in NBOCA). This may reflect the influence of a clinical trial of short course radiotherapy [38], which was recruiting at the time.

The distribution of socio-demographic characteristics of full consenters was broadly similar to reduced consenters (Table 2). The only significant difference was age, where full consenters were significantly more likely to be younger. There was significantly more missing data for ethnicity for full consenters. Full and reduced consenters were generally similar in terms of clinical and treatment characteristics (Table 3), the only significant difference being more screen-detected cases in full consenters (23.0%) compared with the reduced consenters (9.0%).

Table 2 and Table 3 present the distributions of characteristics by participation status at 5 years. Table 4 shows significantly different (p<0.05) baseline characteristics of those who returned a questionnaire at five years and those who did not. Excluding those who were deceased or did not participate at baseline, people who did not return a five year questionnaire were significantly more likely to be ≥80 years, of non-White British ethnicity, to not own accommodation (e.g. renting) and report clinical levels of depression (CES-D> = 20) at baseline.

## Quality of Life

Summary statistics for the QLACS-GSS at each time point is presented in Fig 3. There was a statistically significant decrease in the mean score, indicating QOL improved, at 15 months post-surgery compared to the mean scores observed from baseline to 9 months post-surgery. The difference was more than 10% when comparing the mean scores at 3 months and 9 months with 15 months, suggesting a clinically significant improvement. From 15 months to 5 years, there was little change in the mean QLACS-GSS score.

A similar pattern of results was obtained comparing those with colon cancer and rectal cancer. In addition, for people with rectal cancer, there was a >10% increase on the mean QLCAS-GSS score between baseline and 3 months, suggesting a clinically significant reduction in QOL. At each time point, the mean QLACS-GSS score was slightly higher for those with rectal compared to colon cancer (indicating lower QOL), but this difference was only statistically significant at 3, 24 and 36 months.

605 participants completed the QLACS at baseline and at least one other time point. During the first 18 months following surgery, the proportion with improved QOL increased and the proportion with worsened QOL declined (Fig 4). This pattern then reversed between 18 and 42 months before levelling out over the subsequent months. The proportion whose QOL worsened compared with baseline peaked at 3 months, when 41.8% were in this group, and troughed at 15 months with 25.9% in the group.

The same pattern was seen comparing those with colon and rectal cancer. However, at all time points, there was a greater proportion of people, on average 8.6 percentage points, with rectal cancer who had worsened QOL (data not shown).

**Table 2. Baseline socio-demographic characteristics of approached patients by consent status and of recruited patients by participation status at 5 year follow-up.**

| Baseline socio-demographic characteristics | Type of consent at baseline (N = 1006 consented) | | | Returned five-year questionnaire? (N = 617 completed baseline) | | |
|---|---|---|---|---|---|---|
| | Reduced consent (n = 145) | Full consent (n = 861) [1] | Sig. | No (N = 263) | Yes (N = 354) | Sig. |
| | n (%) | n (%) | | n (%) | n (%) | |
| **Age (years)** | | | *** | | | *** |
| ≥ 50 | 4 (2.8) | 58 (6.7) | | 20 (7.6) | 23 (6.5) | |
| 51–60 | 8 (5.5) | 124 (14.4) | | 42 (16) | 59 (16.7) | |
| 61–70 | 43 (29.7) | 316 (36.7) | | 82 (31.2) | 169 (47.7) | |
| 71–80 | 55 (37.9) | 263 (30.5) | | 84 (31.9) | 87 (24.6) | |
| 81 + | 34 (23.5) | 98 (11.4) | | 35 (13.3) | 16 (4.5) | |
| Missing | 1 (0.7) | 2 (0.2) | | n/a | n/a | |
| **Gender** | | | | | | |
| Male | 79 (54.5) | 513 (59.6) | | 151 (57.4) | 207 (58.5) | |
| Female | 64 (44.1) | 348 (40.4) | | 112 (42.6) | 147 (41.5) | |
| Missing | 2 (1.4) | 0 - | | n/a | n/a | |
| **Ethnicity** | | | ** | | | ** |
| White British | 124 (85.5) | 637 (74.0) | | 216 (82.1) | 298 (84.2) | |
| Other ethnic group | 9 (6.2) | 50 (5.8) | | 27 (10.3) | 16 (4.5) | |
| Unknown ethnicity | 12 (8.3) | 174 (20.2) | | 20 (7.6) | 40 (11.3) | |
| **Deprivation index** | | | | | | |
| 1st quintile (least deprived) | 22 (15.2) | 171 (19.9) | | 44 (16.7) | 71 (20.1) | |
| 2nd quintile | 29 (20.0) | 169 (19.6) | | 47 (17.9) | 81 (22.9) | |
| 3rd quintile | 21 (14.5) | 163 (18.9) | | 44 (16.7) | 68 (19.2) | |
| 4th quintile | 24 (16.6) | 162 (18.8) | | 55 (20.9) | 67 (18.9) | |
| 5th quintile (most deprived) | 29 (20.0) | 174 (20.2) | | 64 (24.3) | 64 (18.1) | |
| Missing | 20 (13.8) | 22 (2.6) | | 9 (3.4) | 3 (0.9) | |
| **Domestic status** | | | n/a | | | |
| Married | n/a | 495 (57.5) | | 164 (62.4) | 242 (68.4) | |
| Living with partner | n/a | 39 (4.5) | | 22 (8.4) | 13 (3.7) | |
| Widowed | n/a | 110 (12.8) | | 34 (12.9) | 38 (10.7) | |
| Divorced/separated | n/a | 64 (7.4) | | 22 (8.4) | 38 (10.7) | |
| Single | n/a | 44 (5.1) | | 19 (7.2) | 21 (5.9) | |
| Missing | n/a | 109 (12.7) | | 2 (0.8) | 2 (0.6) | |
| **Employment status** | | | n/a | | | |
| Employed | n/a | 202 (23.4) | | 71 (27.0) | 111 (31.4) | |
| Unemployed | n/a | 34 (4.0) | | 12 (4.6) | 18 (5.1) | |
| Retired | n/a | 513 (59.6) | | 178 (67.7) | 222 (62.6) | |
| Missing | n/a | 112 (13.0) | | 2 (0.8) | 3 (0.9) | |
| **Accommodation status** | | | n/a | | | ** |
| Owner-occupier | n/a | 596 (69.2) | | 190 (72.2) | 299 (84.5) | |
| Renting | n/a | 126 (14.6) | | 58 (22.1) | 44 (12.4) | |
| Other | n/a | 29 (3.4) | | 13 (4.9) | 8 (2.3) | |
| Missing | n/a | 110 (12.8) | | 2 (0.8) | 3 (0.8) | |

*** Significant difference was found using a Chi-square test (p<0.001) excluding missing.

** Significant difference was found using a Chi-square test (p<0.01) excluding missing.

[1] excluding 10 full consent patients who withdrew at baseline prior to data collection and one who died.

*Definitions*: n/a = not applicable.

**Table 3. Baseline clinical and treatment characteristics of approached patients by consent status and of recruited patients by participation status at 5 year follow-up.**

| Baseline clinical & treatment characteristics | Type of consent at baseline (N = 997 consented) | | | | Returned five-year questionnaire? (N = 617 completed baseline) | | | |
|---|---|---|---|---|---|---|---|---|
| | Reduced consent (n = 145) | | Full consent (n = 852) [1] | | No (N = 263) | | Yes (N = 354) | |
| | n (%) | | n (%) | | n (%) | | n (%) | |
| **Body mass index (kg/m$^2$)** | | | | | | | | |
| Normal/pre-obese (bmi <29.9) | 84 | (58.0) | 436 | (51.2) | 133 | (50.6) | 188 | (53.1) |
| Overweight (bmi> = 30.0) | 23 | (15.8) | 157 | (18.4) | 57 | (21.7) | 71 | (20.1) |
| Unknown BMI | 38 | (26.2) | 259 | (30.4) | 73 | (27.8) | 95 | (26.8) |
| **How cancer was detected** | | | | | | | | |
| Screen-detected | 13 | (9.0) | 196 | (23.0) | 60 | (22.8) | 102 | (28.8) |
| Emergency surgery | 3 | (2.1) | 38 | (4.5) | 10 | (3.8) | 11 | (3.1) |
| Symptomatic | 123 | (84.8) | 597 | (70.2) | 183 | (69.6) | 233 | (65.8) |
| Other | 1 | (0.7) | 8 | (0.9) | 4 | (1.5) | 2 | (0.6) |
| Missing | 5 | (3.4) | 13 | (1.4) | 6 | (2.3) | 6 | (1.7) |
| **Tumour site** | | | | | | | | |
| Colon | 98 | (67.6) | 550 | (64.6) | 173 | (65.8) | 232 | (65.5) |
| Rectal | 47 | (32.4) | 302 | (35.4) | 87 | (33.1) | 122 | (34.5) |
| Missing | 0 | - | 0 | - | 3 | (1.1) | 0 | (-) |
| **Dukes' stage** | | | | | | | | |
| A | 10 | (6.9) | 120 | (14.1) | 35 | (13.3) | 60 | (17.0) |
| B | 78 | (53.8) | 452 | (53.1) | 154 | (58.6) | 189 | (53.4) |
| C1 | 36 | (24.8) | 170 | (20.0) | 52 | (19.8) | 68 | (19.2) |
| C2 | 21 | (14.5) | 99 | (11.6) | 18 | (6.8) | 31 | (8.8) |
| Could not be determined [+] | 0 | - | 11 | (1.3) | 4 | (1.5) | 6 | (1.7) |
| **Nodal status** | | | | | | | | |
| N0 | 86 | (59.3) | 536 | (62.9) | 171 | (65.0) | 238 | (67.2) |
| N1 | 35 | (24.1) | 167 | (19.6) | 46 | (17.5) | 67 | (18.9) |
| N2 | 19 | (13.1) | 98 | (11.5) | 23 | (8.8) | 32 | (9.0) |
| Missing | 5 | (3.4) | 51 | (6.0) | 23 | (8.7) | 17 | (4.8) |
| **Extramural vascular invasion** | | | | | | | | |
| No | 91 | (62.8) | 618 | (72.5) | 196 | (74.5) | 272 | (76.8) |
| Yes | 44 | (30.3) | 191 | (22.4) | 48 | (18.3) | 64 | (18.1) |
| Missing | 10 | (6.9) | 43 | (5.0) | 19 | (7.2) | 18 | (5.1) |
| **Primary surgery type** | | | | | | | | |
| Laparoscopic | 77 | (53.1) | 468 | (54.8) | 150 | (57.0) | 199 | (56.2) |
| Open surgery | 60 | (41.4) | 341 | (40.1) | 100 | (38.0) | 136 | (38.4) |
| Missing | 8 | (5.5) | 43 | (5.0) | 13 | (4.9) | 19 | (5.4) |
| Primary surgical resection | | | | | | | | |
| R0 (complete) | 125 | (86.2) | 739 | (86.7) | 233 | (88.6) | 311 | (87.9) |
| R1 (microscopic residual tumour) | 11 | (7.6) | 37 | (4.3) | 5 | (1.9) | 11 | (3.1) |
| Missing | 9 | (6.2) | 76 | (8.9) | 25 | (9.5) | 32 | (9.0) |
| **Neo-adjuvant treatment** | | | | | | | | |
| No | 114 | (78.6) | 690 | (81.0) | 213 | (81.0) | 288 | (81.4) |
| Yes | 31 | (21.4) | 157 | (18.4) | 43 | (16.3) | 66 | (18.6) |
| Missing | 0 | - | 5 | (0.6) | 7 | (2.7) | 0 | (-) |
| *Among Yes:* | | | | | | | | |
| *- Chemotherapy only* | *1* | *(12.1)* | *19* | *(12.1)* | *5* | *(1.9)* | *10* | *(2.8)* |

*(Continued)*

**Table 3.** (Continued)

| Baseline clinical & treatment characteristics | Type of consent at baseline (N = 997 consented) | | Returned five-year questionnaire? (N = 617 completed baseline) | |
|---|---|---|---|---|
| | Reduced consent (n = 145) | Full consent (n = 852) [1] | No (N = 263) | Yes (N = 354) |
| | n (%) | n (%) | n (%) | n (%) |
| *- Radiotherapy only* | 12 (43.3) | 68 (43.3) | 17 (6.5) | 29 (8.2) |
| *- both* | 18 (44.6) | 70 (44.6) | 21 (8.0) | 27 (7.6) |
| **Adjuvant treatment** | | | | |
| No | 102 (70.3) | 556 (65.3) | 183 (69.6) | 223 (63.0) |
| Yes | 43 (29.7) | 296 (34.7) | 77 (29.3) | 131 (37.0) |
| Missing | 0 - | 0 - | 3 (1.1) | 0 (-) |
| *Among Yes:* | | | | |
| *- Chemotherapy only* | 41 (95.3) | 278 (93.9) | 74 (28.1) | 126 (35.6) |
| *- Radiotherapy only* | 0 - | 6 (2.0) | 1 (0.4) | 1 (0.3) |
| *- both* | 2 (4.7) | 12 (4.1) | 2 (0.8) | 4 (1.1) |
| **Biological therapy** | | | | |
| No | 125 (86.2) | 702 (82.4) | 216 (82.1) | 303 (85.6) |
| Yes | 7 (4.8) | 84 (9.9) | 16 (6.1) | 30 (8.5) |
| Missing | 13 (9.0) | 66 (7.7) | 31 (11.8) | 21 (5.9) |
| *Among Yes:* | | | | |
| *- Cetuximab* | 0 - | 10 (11.9) | 2 (0.8) | 3 (0.9) |
| *- Avastin* | 6 (85.7) | 65 (77.4) | 13 (4.9) | 23 (6.5) |
| *- Cetuximab & Avastin* | 0 - | 3 (3.6) | 0 - | 0 - |
| *- Panitumumab* | 1 (14.3) | 3 (3.6) | 0 - | 3 (0.9) |
| *- Other* | 0 - | 3 (3.6) | 1 (0.4) | 1 (0.3) |
| **Stoma** | | | | |
| No | 86 (59.3) | 538 (62.8) | 164 (62.4) | 238 (67.2) |
| Yes | 56 (38.6) | 302 (35.2) | 93 (35.4) | 113 (31.9) |
| Missing | 3 (2.1) | 17 (2.0) | 6 (2.3) | 3 (0.9) |
| *Among Yes:* | | | | |
| *- Temporary* | 24 (42.9) | 182 (60.3) | 65 (24.7) | 68 (19.2) |
| *- Permanent* | 23 (41.1) | 92 (30.5) | 23 (8.8) | 36 (10.2) |
| *- Duration unknown* | 9 (16.1) | 28 (9.3) | 5 (1.9) | 9 (2.5) |

None of the above characteristics were significantly different at the 5% level (Chi-square test, excluding missing), with one exception of the cancer detection route by the type of consent (p<0.01)

[1] excluding 15 full consent patients who withdrew at baseline prior to data collection and five patients who did not consent to collection of medical details

[+] Dukes stage could not be determined for 11 full consent patients with small tumours following neo-adjuvant therapy.

Statistically significant predictors of worsened QOL in either Model 1 (baseline to two years) or Model 2 (two to five years) are shown in Table 5. Tumour type (colon vs. rectal) was not a significant factor in either model.

## Baseline (pre-surgery) predictors of worsened QOL up two years post-surgery

Compared with people ≥ 60 years, the odds of those aged 71–80 years experiencing worse QOL up to two years were significantly less likely (51% lower), whilst there was no difference for people ≥81 years or for those aged 61–70 years. The odds of worsened QOL over time

**Table 4. Baseline predictors from multivariable logistic regression model of five-year questionnaire withdrawal.**

| Covariates | OR (95% CI) |
|---|---|
| Age groups (ref: 60 or younger) | |
| 61–70 | 0.76 (0.49; 1.18) |
| 71–80 | 1.41 (0.88; 2.26) |
| 81+ | 3.13* (1.54; 6.37) |
| Ethnicity group (ref: White British) | |
| Other | 2.13* (1.08; 4.20) |
| Unknown | 0.77 (0.42; 1.39) |
| Accommodation (ref: owner) | |
| Renting/other | 1.71** (1.20; 2.43) |
| Depression (ref: No, CES-D<20) | |
| Yes, clinically depressed (CES-D> = 20) | 2.09*** (1.38; 3.17) |
| Observations N | 612 |

* p < .05
** p < .01
*** p < .001

were significantly higher among those individuals who had at least two comorbidities (56% higher) and receiving neo-adjuvant treatment (2.4 times higher). People in the 3rd quintile for the deprivation index had nearly twice the odds of worse QOL compared with those in the least deprived quintile. Several psychosocial factors at baseline also predicted the likelihood of worsened QOL up to two years: low self-efficacy (SEMCD score <5), inadequate levels of social support (MOS<100), higher negative and lower positive affect scores. Thus, the odds of having worsened QOL over the first two years of follow-up were significantly lower among participants with at least moderate confidence (64% lower) and full social support (32% lower). With one unit increase in the Positive and Negative Affect Schedule (PANAS) negative affect or PANAS positive affect at baseline, the odds of reporting worsened QOL on average increased by 8% or decreased by 3%, respectively.

## Two year post-surgery predictors of worsened QOL up to five years post-surgery

Several factors at 2 years predicted a subsequent decline in QOL over the next three years (see Table 5). Older age became a significant risk factor for worsened QOL, where the odds of reporting poorer QOL at later time points was nearly three times as high for people aged ≥81 years than for those ≤60 years. People who had received adjuvant therapy had 39% lower odds of having later worsened QOL. For psychosocial factors, both the negative and positive scores on the PANAS were associated with subsequent QOL: with one unit increase in PANAS negative affect or PANAS positive affect, the odds of reporting worsened QOL on average increased by 17% or decreased by 9%, respectively. People who had clinical levels of depression at 24 months had on average 76% lower odds of experiencing later worsened QOL. Participants who scored above the threshold for clinical importance (TCI) on the fatigue scale of the QLQ-C30 were twice as likely to have worsened QOL. Scoring above the TCI on the QLQ-C30 cognitive functioning scale increased the odds of worsened QOL by 63%. Finally, there was a significantly increased odds ratio (1.02) of worsened QOL for those people who reported a higher (worse) QLACS cancer-specific score at 24 months since surgery.

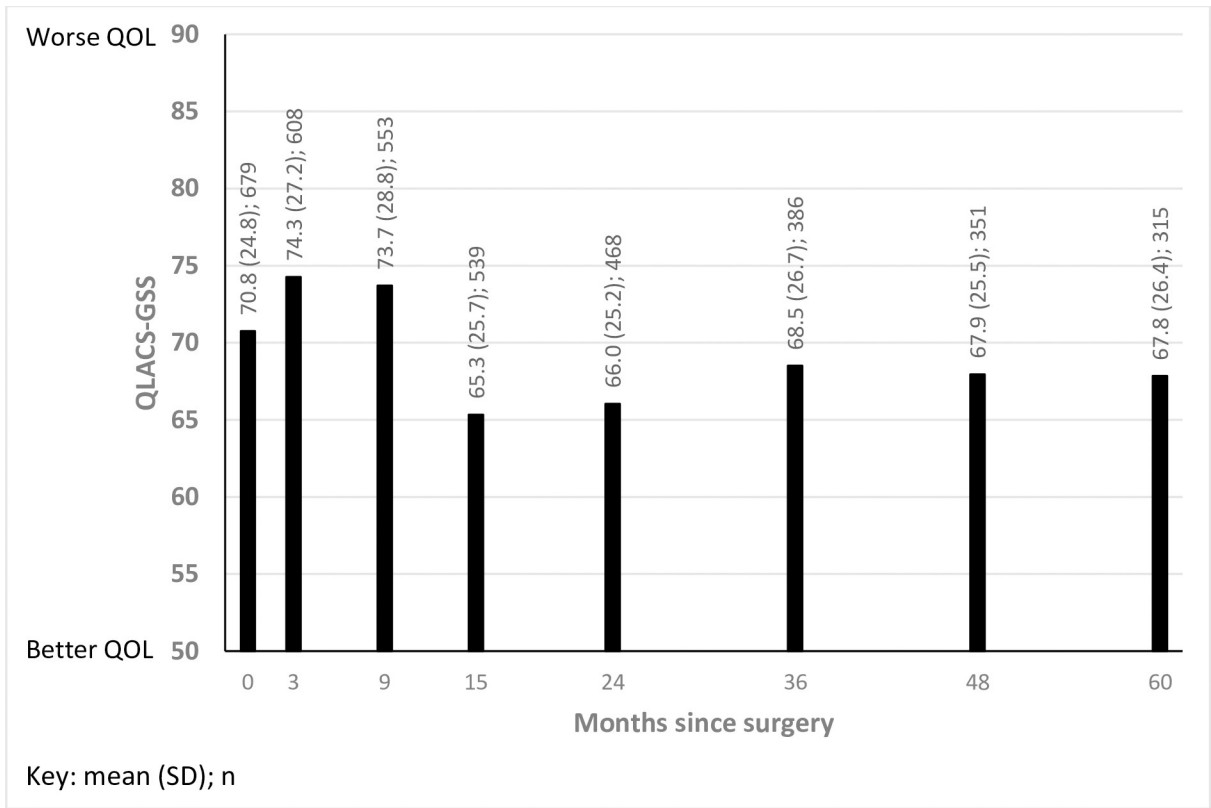

**Fig 3. Mean scores (and standard deviations) of the QLACS-GSS.**

## Sensitivity analyses

Although tumour type was not a significant predictor in either model, we re-ran both models for people with colon cancer and rectal cancer separately. The variables included in the models are listed in supporting information along with the statistically significant predictors. Significant predictors which were associated with worsened QOL in these analyses but not in the main analyses were, for people with colon cancer, obesity in both Model 1 and Model 2, presence of a stoma in Model 1 and frequent urination in Model 2. For people with rectal cancer, life events and high anxiety level were both significant predictors of worsened QOL in Model 2.

## Discussion

CREW is the first large-scale prospective cohort study to capture self-report data on the impact of CRC on people's lives from pre-surgery up to 5 years later. Although previous literature has suggested that most CRC patients experience reasonable to good quality of life (QOL) [3, 10], by using a pre-surgery baseline, this study has demonstrated that around one third of CRC patients have not returned to pre-surgery levels of QOL five years after treatment.

The CREW cohort is a highly representative sample, with a minimum dataset available on 91% of all eligible people during the recruitment period. In general, those who consented to complete questionnaires were representative of all eligible patients, although slightly younger and with earlier stage disease. It is recognised that the most vulnerable people facing CRC may be the most likely to decline study participation due to burden of questionnaire completion

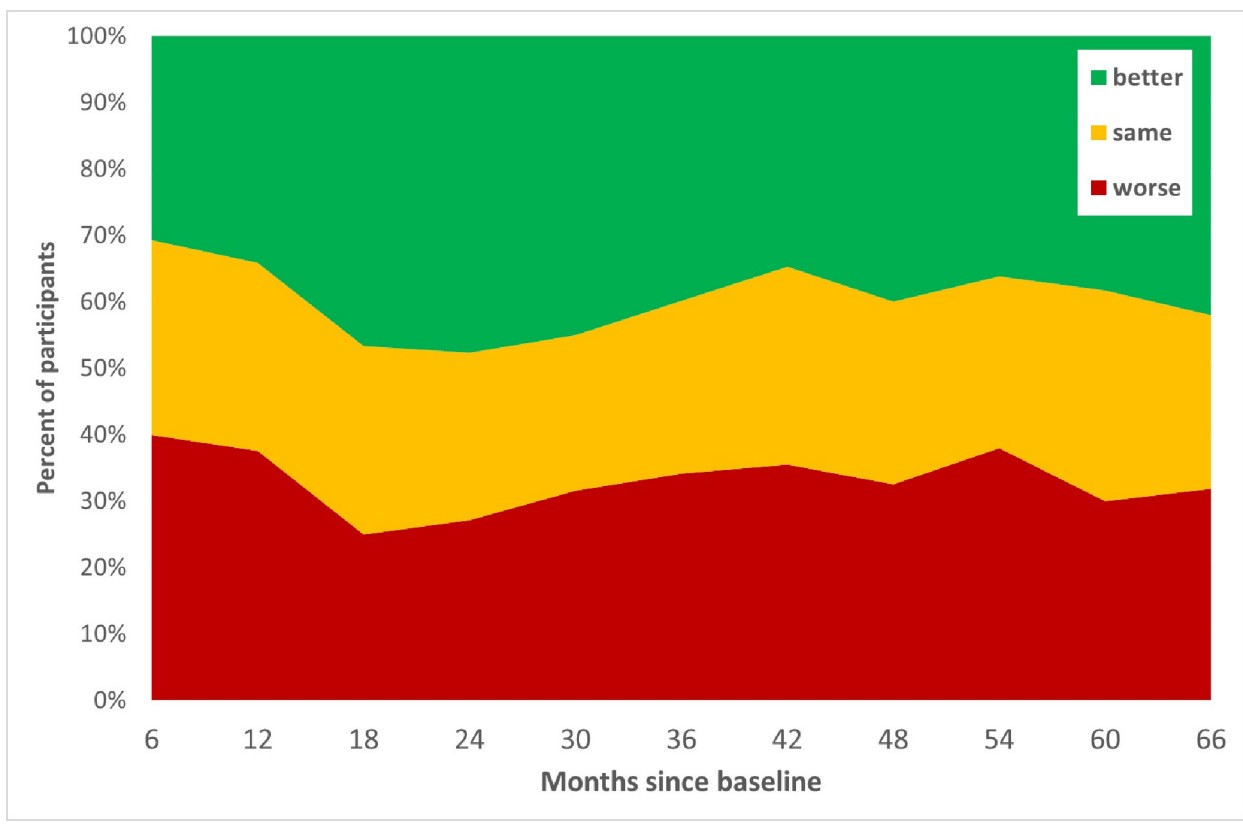

**Fig 4. Changes in quality of life (QLACS-GSS) compared to baseline.** The timeline on the horizontal represents the actual time since surgery (in months) when respondents completed questionnaires.

but our inclusion of a reduced level of consent, where we collected demographic, clinical and treatment details from this group, without the self-report questionnaires, enabled us to collect data on as many eligible patients as possible. Although the full consenters were on average younger than those who gave reduced consent and who declined, older patients are nevertheless still well-represented, with 12% >80 years in the full consenters at baseline. This is important because even though older people represent the majority of cancer patients, they are often under-represented in clinical research [39] and age may represent a risk factor for poor recovery.

Attrition in cohort studies is inevitable, but response rates remained consistently high. Those with depression, aged ≥80 years, of non-White British ethnicity, who did not own their own home were more likely to have stopped participating. These characteristics are often associated with participants who do not continue in longitudinal studies [40]. It is possible that our analyses may underestimate difficulties and need. Nevertheless, those who continued to return questionnaires were broadly representative of the baseline cohort. With the high retention rate and representativeness of the CREW cohort, along with the fact that the baseline measure was intended to be pre-surgery, confidence in the QOL results is high.

## Quality of life

There was a marked improvement in QOL at 15 months, when considering the whole CREW sample, with little change after that up to 5 years. However, in the era of personalised care [41], when the importance of the individual's experience and needs is emphasised, we need to

**Table 5. Multivariable logistic regression models with odds ratios of worsened QOL (ref: improved/same QOL) with statistically significant factors taken separately from two time points.**

| Covariates | Model 1: Baseline | Model 2: 24 m |
|---|---|---|
| | OR (95% CI) | OR (95% CI) |
| *Age groups (ref: 60 or younger)* | | |
| 61–70 | 0.85 (0.57; 1.25) | 1.31 (0.75; 2.28) |
| 71–80 | 0.49** (0.32; 0.75) | 1.63 (0.90; 2.93) |
| 81+ | 0.88 (0.43; 1.83) | 2.88* (1.27; 6.56) |
| *Deprivation index (ref: Least deprived - 1st quintile)* | | |
| 2nd quintile | 0.94 (0.59; 1.49) | |
| 3rd quintile | 2.00** (1.22; 3.29) | |
| 4th quintile | 0.97 (0.61; 1.54) | |
| Most deprived - 5th quintile | 1.45 (0.88; 2.40) | |
| *Neo-adjuvant therapy (ref: none)* | 2.41*** (1.64; 3.53) | |
| *Adjuvant therapy (ref: none)* | N/A | 0.61* (0.41; 0.93) |
| *Comorbidities (ref: none)* | | |
| one | 1.16 (0.77; 1.76) | |
| 2+ | 1.56* (1.05; 2.31) | |
| *PANAS negative score* | 1.08** (1.03; 1.15) | 1.17** (1.04; 1.32) |
| *PANAS positive score* | 0.97* (0.93; 0.99) | 0.91*** (0.86; 0.96) |
| *Full social support MOS = 100 (ref: <100)* | 0.68* (0.46; 0.99) | |
| *QLACS Cancer-specific score* | N/A | 1.02* (1.003; 1.04) |
| *Depression CES-D> = 20 (ref: <20)* | | 0.24** (0.09; 0.62) |
| *Self-efficacy (ref: low confidence)* | | |
| - moderate confidence | 0.36** (0.19; 0.70) | |
| - confident | 0.45* (0.24; 0.86) | |
| - very confident | 0.34** (0.16; 0.74) | |
| *Scoring above threshold for clinical importance on QLQ-C30 fatigue scale (ref: not)* | | 2.03* (1.08; 3.78) |
| *Scoring above threshold for clinical importance on QLQ-C30 cognitive functioning scale (ref: not)* | | 1.63* (1.02; 2.60) |
| *N observations (N individuals)* | 1,767 (524) | 870 (342) |

* p < .05

** p < .01

*** p < .001; N/A = not available at this time-point; empty cells indicate statistical insignificance (p> = .05); the wave of participation was statistically significant in each model (not presented); the original QLACS-GSS score was accounted for to control for the floor and ceiling effects, which was significant in each model (not presented).

consider changes to QOL with respect to baseline scores. Baseline was pre-surgery but post-diagnosis, so it may be that QOL at this time was different to pre-morbid levels. Nevertheless, baseline scores provide a useful benchmark for interpreting changes in QOL scores for individuals. We encourage CRC services and interventions to evaluate efficacy by adopting the practice of measuring <u>changes</u> in QOL scores at the individual level, to assess the number of patients whose QOL changes, rather than using mean scores for the whole patient group.

Assuming that the desired outcome would be to have the same or better QOL compared with baseline, between 58 and 74% of participants reached this standard at each time point during follow up, with the peak at 15 months. Conversely, 60% had worse QOL on at least one occasion during follow up and about 30% had worse QOL at 5 years. Whilst it might be expected that some people's QOL declines because of age-related factors, the baseline multivariable logistic regression model indicates that age is not significantly associated with worsened QOL at later time points. The failure of people to return to pre-surgery levels of QOL is therefore not simply because they are getting older.

We identified the risk factors for people whose QOL is likely to get worse, regardless of the current level, at two key points along the patient pathway: after a cancer diagnosis (before CRC surgery) and two years later, after which people with CRC typically have a break in their oncology appointments until their five year surveillance appointment. This is useful information for health care professionals (HCPs) to help identify people who may need extra support and has the potential to help guide appropriate interventions. When HCPs see a person with cancer, they can identify those who already have poor QOL and require immediate support, as well as identify those at risk of a decline in QOL and offer early intervention for those patients.

Our analyses suggested that although QOL for people with rectal cancer tended to be slightly lower than that for people with colon cancer, and more people with rectal cancer experienced worsened QOL, the factors which predicted worsened QOL were similar for both over time. Some clinical factors predicted worsened QOL during recovery including neo-adjuvant and adjuvant treatment, and for people with colon cancer, presence of a stoma. We did not find an association between surgery type (open or laparoscopic) and worsened QOL during recovery, and we did not have the data to explore the impact of other peri- and the immediate post-operative factors. Whilst HCPs are likely to be familiar with the clinical risk factors for reduced QOL following CRC surgery, our study highlights the importance of psychosocial factors. These are important because they may be amenable to change, with the possibility of implementing interventions to address psychosocial factors from diagnosis.

Around the time of diagnosis, factors which were associated with worsened QOL at later time points, and are potentially amenable to change in the whole cohort, were low levels of social support, low levels of self-confidence, high negative affect and low positive affect. These findings were consistent with the extant literature identifying links between absolute QOL scores and self-efficacy, social support and comorbidities [42, 43]. People who are experiencing these problems could be offered early support in these specific areas which may also help maintain or improve later QOL. Having two or more comorbidities was also predictive of worsening QOL at baseline. Whilst having comorbidities *per se* may not be amenable to change, the way comorbidities are managed could be. Our previous work has shown that it is the impact of comorbidities on daily life which influences QOL [44], and personalised assessment and management for people with comorbidities may improve outcomes.

The social deprivation results were unexpected: whilst the middle (3rd quintile) had twice the odds of experiencing worsened QOL in the first two years compared with those in the least deprived quintile (1st quintile), there was no statistically significant difference between those in the 1st quintile and either the 4th or the 5th quintile. This may be an artefact, perhaps due to the limitations of the assessment of social deprivation (through postcodes).

It is perhaps also unexpected that people with clinical levels of depression two years after surgery were less likely to experience worsened QOL in the next three years. One explanation for this is that people with clinical levels of depression will already have poor QOL, meaning they are less likely to get even worse [45]. From a clinical perspective, anyone experiencing clinical levels of depression should be flagged as in need of immediate further support. Less unexpected was the finding that those with low positive affect and high negative affect at two years were more likely to experience subsequent worsened QOL. This could suggest a link between positive outlook and outcome or it could reflect the overlap between affect and QOL.

Symptoms and functioning were not assessed pre-surgery, but two years later, high levels of fatigue and poor cognitive functioning were associated with later worsened QOL. Although levels of fatigue and cognitive functioning may not be amenable to change, interventions to improve self-management of these may be possible. For example, our research group has developed a web-based resource (RESTORE, freely available at www.macmillanrestore.org.uk) which has been demonstrated to be beneficial to people with cancer-related fatigue [46].

The way age acted as a significant risk factor pre-surgery and two years post-surgery changed. Pre-surgery, those who were aged 71–80 years were less likely to experience worsening QOL compared to other age groups up to two years, and there was no difference between the very youngest and very oldest participants. Whereas at two years post-surgery, those aged ≥71 years were more likely to experience worse QOL over the next three years. The relationship between age and QOL may not be straightforward partly because there is a change in the way people define good QOL as they age [47]. In addition, previous research has shown that elderly people who had experienced CRC rated their global HRQOL as better than or comparable to age-matched controls from the general population up to 10 years after treatment [12]. This "positive outlook" means that it is really important to intervene when an elderly person experiences a decline in QOL, as defined using their own previous scores.

Finally, some additional factors were identified as predictive of worsened QOL when we looked at colon and rectal patients separately–obesity, urinary frequency and stoma for people with colon cancer and anxiety and life events for people with rectal cancer. It is not clear why these factors were only identified in the specific tumour groups since, for example, there is already strong evidence of the link between high BMI and low QOL in people with CRC [48]. Similarly, the link between anxiety and reduced QOL has also been demonstrated in a mixed CRC sample [45].

## Conclusion

The CREW cohort dataset is representative, with relatively low attrition rates, and includes prospective patient-reported data from pre-surgery through to five years post-surgery. It is therefore a valuable data set to explore recovery of health and wellbeing following curative intent treatment for CRC, and data access requests are welcomed (see http://www.horizons-hub.org.uk/access_data.html for details).

At least a quarter of people recovering from CRC had worse QOL compared with pre-surgery at each follow up point during the five years post-surgery. Some risk factors for worsening QOL, such as low self-efficacy, social support and management of comorbidities, are amenable to change, and interventions to support this change should be provided, in line with the personalised care agenda Holistic needs assessments in which physical, emotional, practical, financial and spiritual concerns can be discussed with a member of the health and social care team and a care plan developed, are an important component of personalised care. All CRC patients should be offered holistic needs assessments, incorporating the risk factors identified in CREW, from diagnosis. Providers should map services and commission new services where

required to meet the demand generated by holistic needs assessments and subsequent referrals.

## Supporting information

**S1 Table. Validated measures included at each time point in CREW.**
(DOCX)

**S2 Table. Variables included in the two regression models for the whole cohort.**
(DOCX)

**S3 Table. Variables included in the regression models for colon and rectal cancer patients separately.**
(DOCX)

**S4 Table. Results of the separate colon and rectal cancer regression models.**
(DOCX)

## Acknowledgments

We thank all CREW study participants and recruiting NHS Trusts; Carol Hill, Kerry Coleman, Bjoern Schukowsky, Christine May (study support); Matthew Breckons, Cassandra Powers, Alex Recio-Saucedo, Bina Nausheen, Ikumi Okamoto, Kim-Chivers Seymour, Joanne Haviland (researchers); Jo Clough, Alison Farmer (research partners). Members of the Study Advisory Committee: Jo Armes, Janis Baird, Andrew Bateman, Nick Beck, Graham Moon, Claire Hulme, Peter Hall, Karen Poole, Susan Restorick-Banks, Paul Roderick, Claire Taylor, Jocelyn Walters, Fran Williams, Lynn Batehup, Jessica Corner, and Deborah Fenlon.

## Author Contributions

**Project administration:** Natalia V. Permyakova.

**Writing – original draft:** Sally Wheelwright, Amy Din, Deborah Fenlon, Alison Richardson, Samantha Sodergren, Peter W. F. Smith, Jane Winter.

**Writing – review & editing:** Sally Wheelwright, Lynn Calman, Claire Foster.

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
