## [Decision Letter · Decision Letter 0]

15 Jan 2020

PONE-D-19-31377

Does quality of life return to pre-treatment levels five years after curative intent surgery for colorectal cancer? Evidence from the ColoREctal Wellbeing (CREW) study

PLOS ONE

Dear Prof Foster,

Thank you for submitting your manuscript to PLOS ONE. After careful consideration, we feel that it has merit but does not fully meet PLOS ONE’s publication criteria as it currently stands. Therefore, we invite you to submit a revised version of the manuscript that addresses the points raised during the review process.

We would appreciate receiving your revised manuscript by Feb 29 2020 11:59PM. To enhance the reproducibility of your results, we recommend that if applicable you deposit your laboratory protocols in protocols.io, where a protocol can be assigned its own identifier (DOI) such that it can be cited independently in the future. For instructions see: http://journals.plos.org/plosone/s/submission-guidelines#loc-laboratory-protocols

We look forward to receiving your revised manuscript.

Kind regards,

John Souglakos, MD, Ph.D

Academic Editor

PLOS ONE

Journal Requirements:

3. We noticed you have some minor occurrence(s) of overlapping text with the following previous publication(s), which needs to be addressed:

https://doi.org/10.1007/s11764-017-0636-x

In your revision ensure you cite all your sources (including your own works), and quote or rephrase any duplicated text outside the Methods section. Further consideration is dependent on these concerns being addressed.

4. Thank you for stating the following in the Competing Interests section:"I have read the journal's policy and one author of this manuscript has the following competing interest: Professor Deborah Fenlon has received an honorarium for teaching from Roche."

Additional Editor Comments:

Thank you for the submission of the manuscript.

The results are interesting and may contribute to the current knowledge in the field.

Some minor revisions are required.

Please replay to reviewer comments and suggestion.

Sincerely

Reviewers' comments:

Reviewer's Responses to Questions

**Comments to the Author**

1. Is the manuscript technically sound, and do the data support the conclusions?

Reviewer #1: Yes

2. Has the statistical analysis been performed appropriately and rigorously? 

Reviewer #1: Yes

3. Have the authors made all data underlying the findings in their manuscript fully available?

Reviewer #1: Yes

4. Is the manuscript presented in an intelligible fashion and written in standard English?

Reviewer #1: Yes

5. Review Comments to the Author

Reviewer #1: In the present very interesting and well-written manuscript, the authors report on the 5-year postoperative course of QOL, as compared to the preoperative status, in a cohort of patients with colorectal cancer (CRC). They have found that at least 25% of them report a worse QOL at all time points of post-surgery follow up. Of the risk factors, aging is weakly associated with worse postoperative QOL, and only among those of 70-80 year-old. Strong preexisting risk factors for worse post-surgery QOL are comorbidities, poor QOL, depression, low levels of self-confidence, and low levels of social support. The authors support the view that some of these factors, as being amenable to change, should be identify by Health Care Professionals at the stage of initial CRC diagnosis for preemptive intervention, support and improvement.

To my opinion some revision is required in order to strengthen the quality of the manuscript:

• The size of the paper is rather lengthy; in particular the sections of “Introduction” and “Discussion”.

• Including colon and rectal cancer in the same entity of CRC does not represent reality. Unlike colon cancer, rectal cancer patients very often receive pre-operative treatment (mostly chemo-radiotherapy), are subjected to an operation with significantly high postoperative morbidity (anastomotic leak), present severe long-term postoperative functional problems (LARS) and some of them carry a permanent stoma. Although the authors report in the section of “Results” that neo-adjuvant treatment is related to worse postoperative QOL, they do not comment the finding in the section of “Discussion”. I would suggest that the impact of rectal cancer surgery on QOL should be reported separately from that of colon cancer, with an emphasis on the LARS syndrome, the anastomotic failure and stoma management, if data are available.

• Similarly it seems worthy that the authors assess the impact of the approach, open or laparoscopic on the post-surgery QOL.

6. PLOS authors have the option to publish the peer review history of their article (what does this mean?). If published, this will include your full peer review and any attached files.

Reviewer #1: No

---

## [Author Response · Author response to Decision Letter 0]

16 Mar 2020

Reviewer Comments

Thank you to the reviewer for their insightful comments.

To my opinion some revision is required in order to strengthen the quality of the manuscript:

• The size of the paper is rather lengthy; in particular the sections of “Introduction” and “Discussion”.

We acknowledge that the paper is quite lengthy, due to the scale of the project. We have edited parts of the manuscript where possible to reduce length.

• Including colon and rectal cancer in the same entity of CRC does not represent reality. Unlike colon cancer, rectal cancer patients very often receive pre-operative treatment (mostly chemo-radiotherapy), are subjected to an operation with significantly high postoperative morbidity (anastomotic leak), present severe long-term postoperative functional problems (LARS) and some of them carry a permanent stoma. Although the authors report in the section of “Results” that neo-adjuvant treatment is related to worse postoperative QOL, they do not comment the finding in the section of “Discussion”. I would suggest that the impact of rectal cancer surgery on QOL should be reported separately from that of colon cancer, with an emphasis on the LARS syndrome, the anastomotic failure and stoma management, if data are available.

• Similarly it seems worthy that the authors assess the impact of the approach, open or laparoscopic on the post-surgery QOL.

Thank you for highlighting the importance of acknowledging the clinical differences between people with colon and rectal cancer. Of course the reviewer is quite correct but our main analyses which investigated which factors were associated with worsened quality of life (QOL) over the five years did not find an effect of tumour site. If we had focussed on QOL in the shorter term, this may have been a significant factor. This may also explain why we did not find a significant effect of some of the other clinical factors which the reviewer highlights e.g. open vs. laproscopic surgery, and we did not have data available for some of the other factors (this limitation has been noted). Nevertheless, we repeated the descriptive and regression model analyses for people with colon and rectal cancer separately and have added this to the manuscript We have also added text to note that although clinical factors are of course important, health care professionals (HCPs) are likely to be aware of these already. Our intention in this paper is to highlight the importance of psychosocial factors, particularly those which are amenable to change, to encourage HCPs to recommend early intervention for these issues e.g. low self-efficacy.

---

## [Editor Report · Decision Letter 1]

23 Mar 2020

Does quality of life return to pre-treatment levels five years after curative intent surgery for colorectal cancer? Evidence from the ColoREctal Wellbeing (CREW) study

PONE-D-19-31377R1

Dear Dr. Foster,

We are pleased to inform you that your manuscript has been judged scientifically suitable for publication and will be formally accepted for publication once it complies with all outstanding technical requirements.

With kind regards,

John Souglakos, MD, Ph.D

Academic Editor

PLOS ONE
---

## [Editor Report · Acceptance letter]

25 Mar 2020

PONE-D-19-31377R1 

Does quality of life return to pre-treatment levels five years after curative intent surgery for colorectal cancer? Evidence from the ColoREctal Wellbeing (CREW) study 

Dear Dr. Foster:

I am pleased to inform you that your manuscript has been deemed suitable for publication in PLOS ONE. Congratulations! Your manuscript is now with our production department. 

With kind regards,

on behalf of

Professor John Souglakos 

Academic Editor

PLOS ONE